# Learning to Rank for Non Independent and Identically Distributed Datasets

## ABSTRACT

With the growing data privacy concerns, federated machine learning algorithms capable of preserving the confidentiality of sensitive information while enabling collaborative model training across decentralized data sources are attracting increasing interest. In this paper, we address the problem of collaboratively learning effective ranking models from non-independently and identically distributed (non-IID) training data owned by distinct search clients. We assume that the learning agents cannot access each other's data, and that the models learned from local datasets might be biased or underperforming due to a skewed distribution of certain document features or query topics in the learning-to-rank training data. Thus, we aim to instill in the local ranking model learned from local data the knowledge from other models to obtain a more robust ranker capable of effectively handling documents and queries underrepresented in the local collection. To achieve this, we explore different methods for merging the ranking models, thus obtaining in each client a model that excels in ranking documents from the local data distribution but also performs well on queries retrieving documents having distributions typical of a partner's node. In particular, our findings suggest that by relying on a linear combination of the local models, we can improve IR models effectiveness by up to +17.92% in NDCG@10 metric (moving from 0.619 to 0.730), and by up to +19.64% in MAP metric (moving from 0.713 to 0.853).

## CCS CONCEPTS

• **Information systems → Combination, fusion and federated search**.

## KEYWORDS

Learning to Rank, Non-IID, Distributed Search

**ACM Reference Format:**
Anonymous Author(s). 2024. Learning to Rank for Non Independent and Identically Distributed Datasets. In *Proceedings of ACM Conference (Conference'17)*. ACM, New York, NY, USA, 10 pages. https://doi.org/10.1145/nnnnnnn.nnnnnnn

## 1 INTRODUCTION

Modern Information Retrieval (IR) systems commonly handle corpora that surpass the capacity of a single machine. Consequently,

they employ a distributed architecture where the document collection is partitioned, and each partition is assigned to a separate machine and indexed separately. In a complementary way, data protection legislation, such as the EU GDPR, increasingly regulate contexts where documents or user interaction data are private and cannot be shared with external entities/organizations.

In both the previous scenarios, search is performed over document-partitioned indexes where each partial index, or *shard*, manages a distinct subset of documents [7, 16, 21]. The allocation of documents to the various shards may be naturally established by data ownership reasons, i.e., the data is private and has to be managed exclusively according to the rules of the owner organization. Alternatively, it may focus on reducing the latency of interactive search by querying in parallel equally-balanced shards, or on globally reducing the computational cost by searching only a few shards for each query (selective search) without sacrificing search accuracy [2, 16].

Recent works have investigated federated learning techniques in the IR context by proposing federated Learning-to-Rank (LTR) algorithms [17] aimed at building a single global ranking model for all the shards without sharing data [30]. While in traditional LTR, the data to train a ranking model is centralized, in federated LTR, the model is trained collaboratively across the clients participating in the federation. Each client locally computes model updates using its own data and sends the model updates (not the raw data) to a central server. The central server aggregates the updates received and redistributes the improved model to the participating clients. Such model update process is repeated until convergence or another stopping criterion is reached.

A factor influencing the performance of federated learning systems, also in federated LTR scenarios, is the presence of skews or biases in data distribution among clients [31]. This bias can manifest in various forms, such as disproportionate representation of certain demographics or preferences within local datasets. For instance, some clients may contribute more data pertaining to specific query types, topics, or user behaviors, skewing the learning process towards those particular patterns and potentially hindering the overall model's ability to generalize effectively. An interesting analysis of this almost unexplored research area of IR is presented in [26], where the authors perform a comprehensive analysis of the impact of non-independently and identically distributed (non-IID) data on federated Learning-to-Rank and observe that models trained federatively on non-IID data exhibit significantly lower effectiveness, and may experience difficulties to converge.

Addressing and mitigating non-IID data biases is thus crucial for ensuring the learning of fair and accurate ranking models across the federated clients. This paper follows one of the research directions for LTR on non-IID data outlined in [26]. Unlike the standard federated learning scenario where a single global model is finally obtained, we aim to preserve the peculiarities of the local models that fit the specific data distribution in the clients. Our technique

learns independently and separately on each client different ranking models on the local and private data. Still, it exploits the outcomes of the other clients to make the local model more robust and effective. Referring to the taxonomy presented in [26, 31], we address non-IID data that are both of Type 1 and 4, where Type 1 is when the conditional probability of observing a label given the same instance changes across clients, while Type 4 refers to the cases where the number of training data varies significantly across different clients.

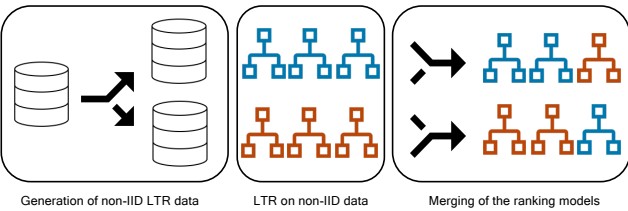

Generation of non-IID LTR data    LTR on non-IID data    Merging of the ranking models

**Figure 1: Scenario under investigation**

The scenario we investigate is visually represented in Figure 1. In this context, we consider $n$ distinct non-IID LTR training datasets, each one employed to train independently a local LTR model. We aim to merge these independently trained LTR models into a distinct and more robust LTR model for each client, capable of handling queries from diverse dataset distributions. Regression tree forests have become a popular choice in Learning to Rank (LTR) due to their ability to model complex relationships between features and rankings. By combining multiple regression trees, these forests can reduce overfitting and improve the overall ranking performance. In our study, we use regression tree forests as the local LTR models, which are merged to create a distinct, more robust, and effective model for each client. Our investigation poses an additional challenge: given the absence of publicly available non-IID LTR dataset, we have first to identify sound strategies for splitting a monolithic LTR dataset into distinct subsets, ensuring the controlled creation of non-IID data of Type 1 and 4.

In light of these challenges, this study aims to address the following research questions:

- RQ1: what strategies can be used to partition a large-scale LTR dataset into Type 1 non-IID data?
- RQ2: In LTR models that use regression tree forests and are trained on non-IID datasets, what methods can be employed to merge these models and enhance the overall predictive power of the final model?

Our experimental results show that the merged models generally surpass the baseline models in performance. Our results on the Istella-S LETOR dataset show that by introducing a linear combination approach, the effectiveness of the IR systems is increased by up to +17.92% in NDCG@10 metric (moving from 0.619 to 0.730), and by up to +19.64% in MAP metric (moving from 0.713 to 0.853). When using a model stacking approach, the effectiveness of the IR systems is increased by up to +16.79% in NDCG@10 metric (moving from 0.619 to 0.723), and by up to +17.39% in MAP metric (moving from 0.713 to 0.837). The rest of the paper is structured as follows. In Section 2 we provide an overview of related work. In Section 3 we introduce the problem of creating non-IID LTR datasets. In Section

4 we discuss the various strategies for the model combination task. Section 5 details the datasets used and the experimental settings. Additionally, we describe the results of the experiments conducted to answer our research questions. Finally, Section 6 summarizes the work and suggests potential future research.

## 2 RELATED WORK

Learning-to-Rank is a vast research area where several machine learning techniques have been proposed to rank the documents matching a query as established by a large supervised training set [17]. Most of these approaches solve the problem starting from query-document representations based on handcrafted features. More recently, new neural approaches have also shown to be effective in solving the task. In contrast with the older ones, some of these techniques exploit the text of both the query and the document directly to extract meaningful features and compute the relevance of the query w.r.t. to a document, e.g., pre-trained transformers [10]. In the following, we present a brief overview of traditional LTR methods based on handcrafted features and then survey the literature on federated LTR.

RankNet [4] leverages a probabilistic ranking framework based on a pairwise approach to train a neural network. The difference between the predicted scores of two different documents is mapped to a probability by mean of the sigmoid function. Hence, using the cross-entropy loss this probability is compared with the ground truth labels, and Stochastic Gradient Descent (SGD) is used to minimize this loss. FRank [25] exploits a generative additive model and substitutes the cross-entropy loss with the fidelity loss, a distance metric adopted in physics, superior to cross-entropy when applied on top of the aforementioned probabilistic framework since 1) has minimum in zero, 2) is bounded in [0, 1]. Neither RankNet or FRank directly optimize a ranking metric, e.g., NDCG, and this discrepancy weakens the power of the model. Since ranking metrics are flat and discontinuous, their optimization within the loss function is troublesome. To overcome this issue, LambdaRank [6] heuristically corrects the RankNet gradients by exploiting the rank position of the document in the overall sorting: it multiplies the RankNet gradient with a term that measure the increase in NDCG when switching the terms, generating the so called $\lambda$-gradients. State-of-the-art LtR models include those based on additive ensembles of regression trees learned by Multiple Additive Regression Trees (MART) [11] and LambdaMart [5, 28] gradient boosting algorithms. LambdaMart [5] combines the successful training methodology provided by $\lambda$-gradients with MART. Currently, ensemble of regression trees are the most effective solution among LTR techniques when dealing with handcrafted features. Since such ranking models are made of hundreds of additive regression trees, the tight constraints on query response time require to trade-off between efficiency and ranking quality [8, 12, 20].

Federated learning is a machine learning technique that enables training models across multiple decentralized clients holding local data samples, without exchanging them. Federated learning has been recently explored in LTR scenarios. When the data is horizontally partitioned, indicating that datasets share the same feature space but differ in samples, standard federated learning techniques can be adapted to LTR. For example, the Party-Adaptive

XGBoost (PAX) approach proposed in [24] introduces a gradient boosting algorithm that employs a party-adaptive histogram aggregation method. The method involves constructing a surrogate representation of the data distribution to determine decision tree splits. Similarly, LightGBM[1], a popular gradient boosting framework, offers various distributed learning algorithms suitable for both horizontal and vertical partitions of the training data. These algorithms reduce communication costs and exploit the two-stage voting mechanism proposed in [23] to further enhance efficiency. Looking beyond traditional horizontal and vertical federated learning approaches, [27] introduces a novel framework called Cross-Silo Federated Learning-to-Rank. The training data in this framework is *cross-partitioned*. Each party collaborates with others to generate training instances, while the documents and queries of each party remain locally stored to ensure privacy protection. To enhance efficiency and protect privacy, data exchange between parties employs a sketching algorithm [13, 22], to compress data in a manner that facilitates query answering. Finally, in [15] a federated version of an online LTR technique is proposed which learns a ranking model from implicit feedback collected from user interactions directly stored on the users' individual devices.

In typical federated learning, models are trained synchronously, exchanging data with a central server to update a global model at each iteration. In contrast, our scenario involves the independent training of models on separate datasets, with aggregation only occurring afterward. To our knowledge, no studies have assessed the effectiveness of LTR models based on forests of decision trees in a non-IID data scenario. While some methods have been proposed to optimally combine two or more rankers, such as the one presented in [29], these methods have not focused on non-IID training datasets.

## 3 GENERATION OF NON-IID LTR DATA

Publicly available LTR datasets are independently and identically distributed (IID) and we need to transform them into non-IID for our experiments. In [26] the authors proposes practical strategies for generating non-IID LTR data of Type 1, 2, 3 and 4 to test their online federated learning solution. They also highlight that Type 1 non-IID data (together with specific cases of Type 2 data) has the most severe effect on the effectiveness of the tested LTR solution. By relying on this result, we also consider Type 1 non-IID data where the conditional probability of observing a label given the same instance changes across clients. For generating Type 1 non-IID data, the authors of [26] consider the search intent simulated for TREC Web Track 2009 to 2012 queries as detailed in [32]. Unfortunately we cannot rely on the same non-IID data due to the small size of these collections, i.e., about 200 queries in total, which does not allow to experiment state-of-the-art LTR solutions requiring many more annotated queries [9, 17].

We thus resort to Istella-S[2], a publicly available large-scale LTR dataset providing among the features modelling each query-document pair the topic-based category of the labelled web document [19]. This dataset comprises $33k$ queries and uses 220 features to represent each one of the $388k$ query-document pairs. The dataset's

[1]https://lightgbm.readthedocs.io/en/latest/Parallel-Learning-Guide.html
[2]https://istella.ai/data/letor-dataset/

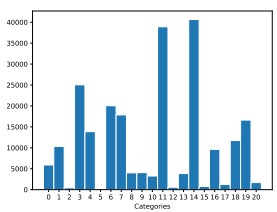
(a) Document distribution

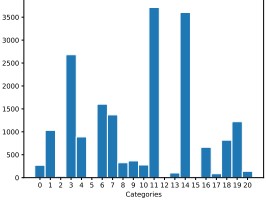
(b) Query distribution

**Figure 2: Category distributions for relevant documents and queries in the Istella training set.**

features are categorized into four main groups: query features, document features, query-document features, and proximity-based features. The dataset is pre-divided into training, validation, and test sets according to a 60%-20%-20% scheme. The feature vectors are encoded in SVM-Rank format[3]. In the dataset, each line represents a query-document pair, featuring the relevance label as the first field (ranging from 0 for not relevant to 4 for perfectly relevant). The second field denotes the query identifier (qid), followed by the features described in the format index:value. LTR models are usually trained list-wise at the query level, and in training datasets, we usually have more relevant and irrelevant documents annotated for the same query. To simulate our distributed LTR non-IID scenario where clients manage documents and answer queries belonging to different topics, we thus need a topic-based category for queries, not documents. In the following we discuss how topic-based labels are assigned to LTR queries of Istella, and how we can exploit these labels to generate Type 1 non-IID partitions of the LTR dataset.

### 3.1 Assigning topic-based labels to LTR queries

We propagate the topic-based label from documents to queries by considering the relevance labels of query-document pairs in the Istella dataset. Given a query $q$ and the set $R_q$ of documents labeled as relevant for $q$, we assign $q$ to the most frequent category among documents in $R_q$. In case of a tie, the selection is based on the category with the highest sum of relevance scores. If there is a further tie, the preference is given to the category containing the document with the highest relevance label. In the unlikely event of an additional tie, the category is assigned randomly from the candidate categories. Figure 2a and 2b plot the distribution of categories for relevant documents and queries in the Istella training set, respectively. The similarity between the two distributions highlights the effectiveness of the proposed category assignment method.

### 3.2 Generating Type 1 splits of the LTR dataset

For the purposes of this work we generate two distinct splits of the Istella dataset of Type 1 non-IID partitions by exploiting the query topic-based categories previously discussed. Specifically, we consider the two most popular categories in the dataset, namely categories 11 and 14 (see Figure 2b). Either the dataset-splitting approach or the methods discussed below for combining the knowledge coming from the local models can be trivially extended to

[3]https://www.cs.cornell.edu/people/tj/svm_light/svm_rank.html

a greater number of clients. The splits are computed as follows: given a dataset $DS$ and a query category $c$, $DS_1$ contains all the training queries belonging to $c$, while $DS_2$ incorporates the remaining queries. We expect the LTR process to obtain two models, $M_1$ trained on $DS_1$ and $M_2$ trained on $DS_2$, with the following properties: $M_1$ performs well on data aligned with the category distribution of $DS_1$, but shows a decrease in effectiveness on data corresponding to the category distribution of $DS_2$, and vice versa for $M_2$. Given the unbalanced distribution of training queries in $DS_1$ and $DS_2$, the non-IID LTR data generated with our procedure are both of Type 1 and Type 4 [26].

## 4 MODEL COMBINATION

We explore two methodologies for merging the models trained independently on each client into an improved local ranking model as illustrated in Figure 1. To this end, we perform the merging of the local ranking models on each client by exploiting a small number of labeled queries from the validation set that are the most dissimilar from the typical distribution observed on the specific client. Hereinafter we will call this dataset used for optimizing the merging $D_{merge}$. The first merging method involves computing a linear combination of the scores computed by the various models, while the second solution requires the learning of a simple stacking model.

*Linear combination of the scores.* The first method, similar to that proposed in [29], linearly combines the scores from multiple rankers such that the resulting score achieves the highest Normalized Discounted Cumulative Gain (NDCG).

Assuming $s_i(q, d)$ represents the output score of local model $M_i$ for query-document pair $(q, d)$, then the output score $s(q, d)$ of the the model $M$ combining the output of all the $n$ local models can be written as:

$$s(q, d) = \sum_{i=1,\ldots,n} \alpha_i \cdot s_i(q, d)$$

where weights $\alpha_i \geq 0$ and $\sum_{i=1,\ldots,n} \alpha_i = 1$, are optimized on $D_{merge}$ using a grid search procedure.

*Model Stacking.* Model stacking involves training a new model by learning from the predictions of the local models. The idea is to leverage the diverse predictions to improve the overall performance.

In our case, the output scores of local models $M_i$ are used as input features for learning a new model $M$, possibly different for each client. In the LTR setting, each document-query pair is represented by a feature vector $x$, comprising $|x|$ features. Leveraging the $n$ already trained models $M_i$, we first predict the query-document score with the local models. Subsequently, we add those predictions as additional features. Specifically, the output score $s_i$ from $M_i$ is added to the features describing query-document pair as feature $f_{|x|+i}$. Finally, a new simple model is trained on this score-augmented version of $D_{merge}$.

## 5 EXPERIMENTAL EVALUATION

This section introduces the experimental settings and our experimental findings regarding both the dataset splitting and the model combination. While our proposed approach can be applied to any

**Table 1: nDCG@10 obtained by the models trained on the initial split**

|          | $DS_1$ | $DS_2$ | DS    |
|----------|--------|--------|-------|
| $M_1$    | 0.761  | 0.728  | 0.741 |
| $M_2$    | 0.747  | 0.771  | 0.786 |
| $M_{full}$ | 0.767 | 0.775  | 0.789 |

number of non-IID datasets, in our experiments, we considered a simplified scenario with two non-IID datasets, for ease of evaluation and demonstration purposes. Despite this simplification, our experiments still provide valuable insights into the effectiveness of our approach in handling non-IID data and its potential for real-world applications.

### 5.1 Experimental Setting

All the experiments leverage the LambdaMART implementation available in the LightGBM gradient boosting library[4] [14]. Noteworthy optimizations in LightGBM include the use of histogram-based algorithms that bin continuous feature values for accelerated training and reduced memory consumption [13, 22].

Hyper-parameter tuning is performed using the Optuna library[5] [1], which uses the Tree-structured Parzen Estimator (TPE) algorithm [3]. On each trial, for each parameter, TPE fits one Gaussian Mixture Model (GMM) $l(x)$ to the set of parameter values associated with the best objective values, and another GMM $g(x)$ to the remaining parameter values. It chooses the parameter value $x$ that maximizes the ratio $l(x)/g(x)$. We optimized four learning parameters: `learning_rate` in [0.0001, 0.0005, 0.001, 0.005, 0.01, 0.05, 0.1, 0.2, 0.3], `num_leaves` in 15,255, `min_data_in_leaf` in 10,500, with a step size of 5, and `min_sum_hessian_in_leaf` in 1,100. The hyper-parameters of the models are tuned on the validation set based on the NDCG@10 metric. We train a maximum of 3000 trees and implement early stopping to prevent overfitting. Specifically, the training process halts if there is no improvement on the validation set for 100 consecutive iterations.

We evaluate the performance of the methods using nDCG@10, MRR@10 and MAP@10.

### 5.2 Experimental Results

We now report and discuss the results obtained in our experiments.

*Dataset Splitting.* Looking at Figure 2b, we notice that the most frequent query category is category 11, representing queries related to "Lifestyle". We thus assign all queries associated with category 11 to split $DS_1$ and move the remaining queries to $DS_2$. The model trained on the full Istella-S dataset is denoted as $M_{full}$, whereas $M_1$ is the model trained on $DS_1$, and $M_2$ is the model trained on $DS_2$. Subsequently, we test the three models on the test sets of $DS$, $DS_1$, and $DS_2$, obtaining the results illustrated in Table 1. From the numbers reported in the table, we can see that, as expected, $M_1$ performs well on the test set with the same category distribution and worse on the one with $DS_2$'s category distribution and

---

[4]https://github.com/microsoft/LightGBM
[5]https://github.com/optuna/optuna

**Table 2: Statistics of the various dataset splits**

| | | | Partitioning without tie-breaking | | | | Exclusive partitioning | | | |
| | | | Category 11 | | Category 14 | | Category 11 | | Category 14 | |
| | | DS | $DS_1$ | $DS_2'$ | $DS_1$ | $DS_2'$ | $DS_1$ | $DS_2'$ | $DS_1$ | $DS_2'$ |
|---|---|---|---|---|---|---|---|---|---|---|
| *Train Set* | # Queries | 19, 245 | 3, 385 | 1, 351 | 3, 243 | 1, 148 | 270 | 1, 351 | 239 | 1, 148 |
| | # Relevant Documents | 230, 676 | 44, 547 | 13, 723 | 40, 655 | 11, 600 | 3, 287 | 13, 723 | 3, 002 | 11, 600 |
| | # Query-Document Pairs | 2, 043, 304 | 399, 487 | 88, 169 | 333, 947 | 78, 025 | 28, 166 | 88, 169 | 23, 642 | 78, 025 |
| *Validation Set* | # Queries | 7, 211 | 1, 031 | 589 | 1, 307 | 555 | 65 | 589 | 64 | 555 |
| | # Relevant Documents | 80, 592 | 13, 063 | 5, 219 | 14, 592 | 5, 120 | 756 | 5, 219 | 732 | 5, 120 |
| | # Query-Document Pairs | 684, 076 | 116, 642 | 32, 649 | 115, 036 | 34, 234 | 6, 742 | 32, 649 | 5, 591 | 34, 234 |
| *Test Set* | # Queries | 6, 562 | 531 | 211 | 612 | 201 | 25 | 211 | 26 | 205 |
| | # Relevant Documents | 76, 956 | 6, 831 | 2, 098 | 8, 141 | 1, 981 | 176 | 2, 098 | 248 | 2, 022 |
| | # Query-Document Pairs | 681, 250 | 59, 632 | 15, 961 | 64, 816 | 16, 722 | 2, 057 | 15, 961 | 2, 594 | 17, 043 |

vice versa $M_2$. Taking $M_{full}$ as the baseline, we observe that $M_1$ experiences some loss in performance, especially on the test sets of $DS$ and $DS_2$, but still generalizes well, while $M_2$ is nearly as good as $M_{full}$. This is likely a consequence of how the splits are computed: even if $M_1$ only encounters queries categorized as 11, these queries likely retrieve documents, both relevant and irrelevant, belonging to other categories. Let us consider the training set, which comprises 19, 245 queries. Only for 1, 939 queries (7%) all relevant documents belonged to the dominant category. There was a tie in 2, 372 queries (12%), with at least two categories having the same number of relevant documents. Moreover, we found that 8, 579 queries (45%) displayed a difference equal to or lower than 3 between the number of relevant documents belonging to the dominant category and the next one. On the other hand, 6, 885 queries (36%) had a difference greater than 3 in this regard.

We explored two additional splitting methods for $DS_1$ to address this matter. In our initial splitting approach we employed a tie-breaking logic for queries featuring two or more dominant categories. Our first alternative method, *partitioning without tie-breaking*, eliminates this tie-breaking logic. Specifically, $DS_1$ only contains queries where the count of relevant documents associated with category 11 exceeds that of other categories. The second method, which we call *exclusive partitioning*, takes a more extreme approach. In this case, we select the queries whose relevant documents all belong to category 11, and we allocate them to $DS_1$. Therefore, $DS_1$ does not contain relevant documents belonging to the other categories.

However, as we allocate less data to $DS_1$, a greater volume of data samples flows into $DS_2$, most likely enhancing its predictive capabilities. Our goal is to artificially worsen the results of $M_2$ so that the two models trained on the non-IID subsets perform well on queries containing documents belonging to the categories they were trained on, but demonstrate lower performance on others. This approach enables us to leverage the strengths of each model during the model merging phase, ultimately achieving optimal results on every query. Therefore, for each $DS_1$ partitioning strategy adopted, we create $DS_2'$, a filtered variant of $DS_2$ obtained by excluding from $DS_2$ all the documents belonging to category 11. We achieve

this by eliminating all queries of $DS_2$ containing those specific query-document pairs. It is noteworthy that $DS_2'$ is identical across each $DS_1$ partitioning method previously introduced. The reason is quite straightforward. Each $DS_1$ partitioning produces a distinct $DS_2$ version. These differ only for queries containing at least one relevant document belonging to category 11, which are all removed from $DS_2'$.

We execute these operations on the two most frequent categories, 11 (Lifestyle) and 14 (News), providing a more comprehensive understanding of the effectiveness of the splitting method. This approach will undoubtedly contribute to a more robust evaluation during the merging phase, as methods will be tested on various split types and categories. Table 2 shows the statistics of the produced splits. When comparing the original dataset (DS) to the partitioned datasets ($DS_1$ and $DS_2'$), we observe a significant reduction in the number of queries, relevant documents, and query-document pairs. For instance, in the train set, the number of queries decreases from 19,245 (DS) to around 3,385 ($DS_1$) and 1,351 ($DS_2'$) for *partitioning without tie-breaking*, and to 270 ($DS_1$) and 1,351 ($DS_2'$) for *exclusive partitioning*. This decrease is also seen in the validation and test sets. This reduction in dataset size is expected, as partitioning involves dividing the original dataset into smaller, more focused subsets. When comparing the two partitioning approaches, we see that *exclusive partitioning* results in even smaller datasets compared to *partitioning without tie-breaking*. This is evident for both categories (11 and 14). *Exclusive partitioning* is a more restrictive approach, where each relevant document can only belong to one category. This leads to a more drastic reduction in dataset size, as some queries or documents may be excluded from the partitioned datasets altogether. In contrast, *partitioning without tie-breaking* allows for some overlap between categories, resulting in larger partitioned datasets. For example, in the train set, the number of queries in Category 11 is 3,385 for *partitioning without tie-breaking*, but only 270 for *exclusive partitioning*. Similarly, the number of query-document pairs in Category 14 is 88,169 for partitioning without tie-breaking but only 28,166 for exclusive partitioning.

We present the results of our experiments in Table 3. Specifically, we compare the performance of three models: $M_1$, trained

**Table 3: nDCG@10 results obtained by the models trained on the alternative splits**

| | Partitioning without tie-breaking | | | | | |
|---|---|---|---|---|---|---|
| | Category 11 | | | Category 14 | | |
| | $DS_1$ | $DS'_2$ | DS | $DS_1$ | $DS'_2$ | DS |
| $M_1$ | 0.764 | 0.730 | 0.740 | 0.737 | 0.733 | 0.745 |
| $M'_2$ | 0.698 | 0.765 | 0.722 | 0.670 | 0.754 | 0.717 |
| $M_{full}$ | 0.765 | 0.801 | 0.786 | 0.760 | 0.791 | 0.786 |
| | Exclusive partitioning | | | | | |
| $M_1$ | 0.741 | 0.604 | 0.543 | 0.747 | 0.296 | 0.334 |
| $M'_2$ | 0.718 | 0.765 | 0.722 | 0.720 | 0.754 | 0.717 |
| $M_{full}$ | 0.753 | 0.801 | 0.786 | 0.779 | 0.792 | 0.786 |

on the $DS_1$ partition; $M'_2$, trained on the $DS'_2$ variant of $DS_2$; and $M_{full}$, the baseline model trained on the full Istella dataset $DS$. We omit the results of the model trained on $DS_2$ as they are expected to be similar to those of the baseline model $M_{full}$. Each of these models is evaluated on three test sets: the test set of $DS_1$, the test set of $DS'_2$, and the test set of the original Istella dataset $DS$. Beginning with category 11, we observe that, in the *partitioning without tie-breaking* approach, the local models exhibit the desired characteristics. Specifically, both $M_1$ and $M'_2$ demonstrate strong performance on their respective test sets, but experience a slight decline in performance on test sets drawn from other category distributions. Notably, their performance on the Istella test set is fairly similar, suggesting that while each model may specialize in certain types of queries, their overall performance is not significantly different. The results for the *exclusive partitioning* approach show a similar pattern. However, $M_1$ appears to underperform on both $DS'_2$ and $DS$, likely due to the lower number of queries it was trained on. This could potentially lead to suboptimal results during the merging phase. Regarding category 14, the results for the *partitioning without tie-breaking* approach are similar to the same approach for category 11. However, the worst results are seen for the *exclusive partitioning* of category 14. Given the lower frequency of Category 14 in $DS$ compared to Category 11, the number of queries exclusively containing relevant documents of Category 14 is very limited. This is evident in the performance of $M_1$ on the test sets belonging to other category distributions. As a result, this partitioning approach was discarded.

Based on these results, we choose three split combinations to employ in our experiments. This selection allows us to test our merging algorithms on multiple instances, providing a comprehensive understanding of the strengths and weaknesses of our methods. In each split, for the sake of simplicity in notations, $DS_2$ corresponds to the alternative version $DS'_2$. We define the first split as $Split_1$, where $DS_1$ is obtained using the *exclusive partitioning* method with category 11. To obtain the Non-IID partitioning, we started with the Istella dataset $DS$ and selected all the queries where all the relevant documents belonged to Category 11. These queries were assigned to $DS_1$. The remaining queries were then assigned to $DS_2$, but only if they did not contain any relevant documents of category 11. The other remaining queries were used for the creation of the datasets used to locally merge the models. These datasets are denoted as

$D_{merge}$. The main goal was to obtain a combined model on each client, using data samples with a different category distribution than the one the local model was trained on. To achieve this, we sorted all the remaining queries by the number of relevant documents of category 11. For $Client_1$, model $M_1$ was trained on $|Q_1|$ queries with many relevant documents of category 11. Therefore, we selected the $\frac{|Q_1|}{2}$ queries with the lowest number of relevant documents belonging to category 11 from the sorted list, and used those as $D_{merge}$ to merge $M_1$ and $M_2$ on $Client_1$. As for $Client_2$, we chose the $\frac{|Q_2|}{2}$ queries with the most relevant documents belonging to category 11 from the sorted list, and used them as $D_{merge}$ to combine the models locally.

The second split, denoted as $Split_2$, was obtained in a similar manner. The key difference is that $DS_1$, in this case, corresponds to the split created using the *partitioning without tie-breaking* method with category 11. The rest of the process remained the same as for the first split. Finally, the last split is $Split_3$, where $DS_1$ is the partitioning created using the *partitioning without tie-breaking* method with category 14. The obtained splits align with our expectations performance-wise and also demonstrate Non-IID characteristics.

*Optimally Combining Two Rankers.* Our next step involves testing the merging methods on the three splits obtained in the previous Section. All experiments are assessed using three evaluation measures: nDCG@10, MAP and MRR@10. As previously mentioned, we have two clients, each training a local model on its private data. Our goal is to combine the two models on each node, in order to improve the robustness of the locally trained models. We adopt the following notation:

- $M_1$ denotes the model trained locally on $DS_1$, the private dataset of $Client_1$.
- $M_2$ denotes the model trained locally on $DS_2$, the private dataset of $Client_2$.
- $M_{12\alpha}$ ($M_{21\alpha}$) represents the model obtained by merging the two local models by exploiting the private data samples in $D_{merge}$ of $Client_1$ ($Client_2$). Note that this model varies depending on the specific split and client.

The baseline for this experiment is established by evaluating models $M_1$ and $M_2$ on the Istella-S test set. Essentially, the merged models must outperform $M_1$ and $M_2$ for the merging approach to be

**Table 4: Results obtained using the linear combination method. Superscripts a, b denote statistical significance differences w.r.t. the specified baseline model(s).**

|  | $Split_1$ | | | $Split_2$ | | | $Split_3$ | | |
|---|---|---|---|---|---|---|---|---|---|
|  | nDCG@10 | MAP | MRR@10 | nDCG@10 | MAP | MRR@10 | nDCG@10 | MAP | MRR@10 |
| $\mathbf{M}_1$ (a) | 0.619 | 0.713 | 0.897 | 0.742 | 0.862 | 0.962 | 0.748 | 0.870 | 0.967 |
| $\mathbf{M}_2$ (b) | 0.727 | 0.854 | 0.968 | 0.727 | 0.854 | 0.967 | 0.717 | 0.834 | 0.961 |
| $\mathbf{M}_{12\alpha}$ | $0.730^{ab}$ | $0.853^{ab}$ | $0.966^{ab}$ | $0.752^{ab}$ | $0.875^{ab}$ | $0.972^{ab}$ | $0.750^{ab}$ | $0.870^{b}$ | $0.969^{b}$ |
| $\mathbf{M}_{21\alpha}$ | $0.719^{ab}$ | $0.833^{ab}$ | $0.955^{ab}$ | $0.749^{ab}$ | $0.869^{ab}$ | $0.967^{a}$ | $0.751^{ab}$ | $0.873^{ab}$ | $0.968^{b}$ |

considered effective. As value for $\alpha$ we sweep from 0 to 1 with a step size of 0.01, and select the value that yields the highest nDCG@10 on the validation set. This fine-tuned hyperparameter will then be employed for ranking the test sets. Note that the higher the $\alpha$, the more weight $M_2$ has in the merged score computation, the lower the $\alpha$, the more weight $M_1$ has.

Table 4 shows the results obtained for each split. Starting with $Split_1$, we analyze the results obtained on Client$_1$. The linear combination algorithm yields an optimal $\alpha$ value of 0.79. The results demonstrate that $M_\alpha$ achieves superior performance compared to all baseline models in terms of the nDCG@10 metric. Notably, this is accomplished despite the significant performance gap between $M_1$ and $M_2$, which can be attributed to the smaller training set of $M_1$. Moving on to Client$_2$, the optimal $\alpha$ value is 0.46. Interestingly, the combined model underperforms compared to $M_2$ in this scenario. This phenomenon can be attributed to the bias in the data used to optimize $\alpha$, which contains queries with many relevant documents from category 11. As a result, the optimization process assigns more importance to $M_1$, which underperforms compared to $M_2$ on the original test set.

Regarding Split$_2$, on Client$_1$ the optimal $\alpha$ value is 0.47. The merged model $M_\alpha$ outperforms all the baseline models in terms of the nDCG@10 metric. On Client$_2$, with an optimal $\alpha$ of 0.15, model $M_\alpha$ outperforms all the baselines.

As for the last split, on Client$_1$, the merged model outperforms all the baselines with an optimal $\alpha$ of 0.57. Finally, for Client$_2$, with an optimal $\alpha$ of 0.14, the merged model $M_\alpha$ outperforms all the baselines.

In summary, this method proves effective, especially in scenarios where both $M_1$ and $M_2$ demonstrate decent performance. However, it's essential to note that the overall results would probably be better if data samples with the same category distribution of Istella's test set were used to compute the optimal $\alpha$.

*Model Stacking.* The baseline for this experiment is established by evaluating models $M_1$ and $M_2$ on the Istella test set. We denote the meta-model trained on the scores of $M_1$ and $M_2$ on Client$_1$ as $M_{12S}$ and the one trained on Client$_2$ as $M_{21S}$. The results obtained using model stacking are presented in Table 5. We begin our analysis with $Split_1$ on Client$_1$, where we perform hyper-parameter tuning on $M_S$. The optimal configuration consists of 60 trees, a learning rate of 0.2, a minimum data in leaf of 285, 143 leaves, and a minimum sum Hessian in leaf of 21. Notably, $M_S$ outperforms $M_1$, but not $M_2$, likely due to $M_1$'s underperformance compared to $M_2$ on the test set. Moving on to Client$_2$, the optimal $M_S$ is configured with

321 trees, a learning rate of 0.05, a minimum data in leaf of 355, 185 leaves, and a minimum sum Hessian in leaf of 3. Although $M_S$ performs slightly better, it is still outperformed by $M_2$.

Continuing with $Split_2$ on Client$_1$, we perform hyper-parameter tuning for $M_S$. The optimized configuration consists of 424 trees, a learning rate of 0.1, a minimum data in leaf of 405, 65 leaves and a minimum sum hessian in leaf of 66. In a scenario where both $M_1$ and $M_2$ exhibit similar performances on the test set, this method improves the result of all the baseline models in every metric. The same results are found on Client$_2$, where the optimal $M_S$ has 132 trees, a learning rate of 0.2, a minimum data in leaf of 190, 93 leaves and a minimum sum hessian in leaf of 90.

Concluding with $Split_3$, on Client$_1$ the tuned model $M_S$ comprises 473 trees, with a learning rate set at 0.05. The optimization process resulted in a minimum data in leaf of 70, 57 leaves, and a minimum sum hessian in leaf of 26. Unfortunately, while beating $M_2$, $M_S$ underperforms with respect to $M_1$. This can be attributed to the higher difference in performance between $M_1$ and $M_2$. However, on Client$_2$, the new model outperforms even $M_1$. The optimal $M_S$ has 139 trees, a learning rate of 0.1, 117 leaves, a minimum data in leaf of 100, and a minimum sum of Hessian in leaf of 40.

Figure 3 presents a feature importance plot for the models trained using the model stacking approach. This plot is a crucial visualization tool in machine learning, as it helps understand the relative contribution of each feature to the predictive power of the model. In the context of model stacking, the features used as input to the meta-model are the output scores of the local models, $M_1$ and $M_2$. These output scores are encoded as features 221 and 222, respectively. The feature importance plot reveals that, across all models, the most important features are indeed those containing the output scores of the local models (features 221 and 222). This suggests that the meta-model is heavily reliant on the predictions made by the local models, $M_1$ and $M_2$. This is a desirable outcome, as it indicates that the model stacking approach is effectively incorporating the knowledge and strengths of the individual models into its learning process. In a Non-IID scenario, the data distribution varies across different domains or partitions, making it challenging for models to generalize well. Despite this challenge, the model stacking approach is able to effectively combine the strengths of the local models, suggesting its robustness in handling Non-IID data.

In summary, our experiments demonstrate that model stacking is an effective approach to improve the performance of base models when their individual performances are comparable. By leveraging the strengths of $M_1$ and $M_2$, which were trained on non-IID

**Table 5: Results obtained using model stacking. Superscripts a, b indicate statistical significance w.r.t. the specified baseline model(s).**

| | $\text{Split}_1$ | | | $\text{Split}_2$ | | | $\text{Split}_3$ | | |
| --- | --- | --- | --- | --- | --- | --- | --- | --- | --- |
| | nDCG@10 | MAP | MRR@10 | nDCG@10 | MAP | MRR@10 | nDCG@10 | MAP | MRR@10 |
| $\mathbf{M}_1$ (a) | 0.619 | 0.713 | 0.897 | 0.742 | 0.862 | 0.962 | 0.748 | 0.870 | 0.967 |
| $\mathbf{M}_2$ (b) | 0.727 | 0.854 | 0.968 | 0.727 | 0.854 | 0.967 | 0.717 | 0.834 | 0.961 |
| $\mathbf{M}_{12S}$ | $0.714^{ab}$ | $0.837^{ab}$ | $0.958^{ab}$ | $0.746^{ab}$ | $0.864^{ab}$ | $0.967^a$ | $0.743^b$ | $0.851^{ab}$ | $0.964^b$ |
| $\mathbf{M}_{21S}$ | $0.723^{ab}$ | $0.833^{ab}$ | $0.954^{ab}$ | $0.746^{ab}$ | $0.859^{ab}$ | $0.962^b$ | $0.749^b$ | $0.868^{ab}$ | $0.968^b$ |

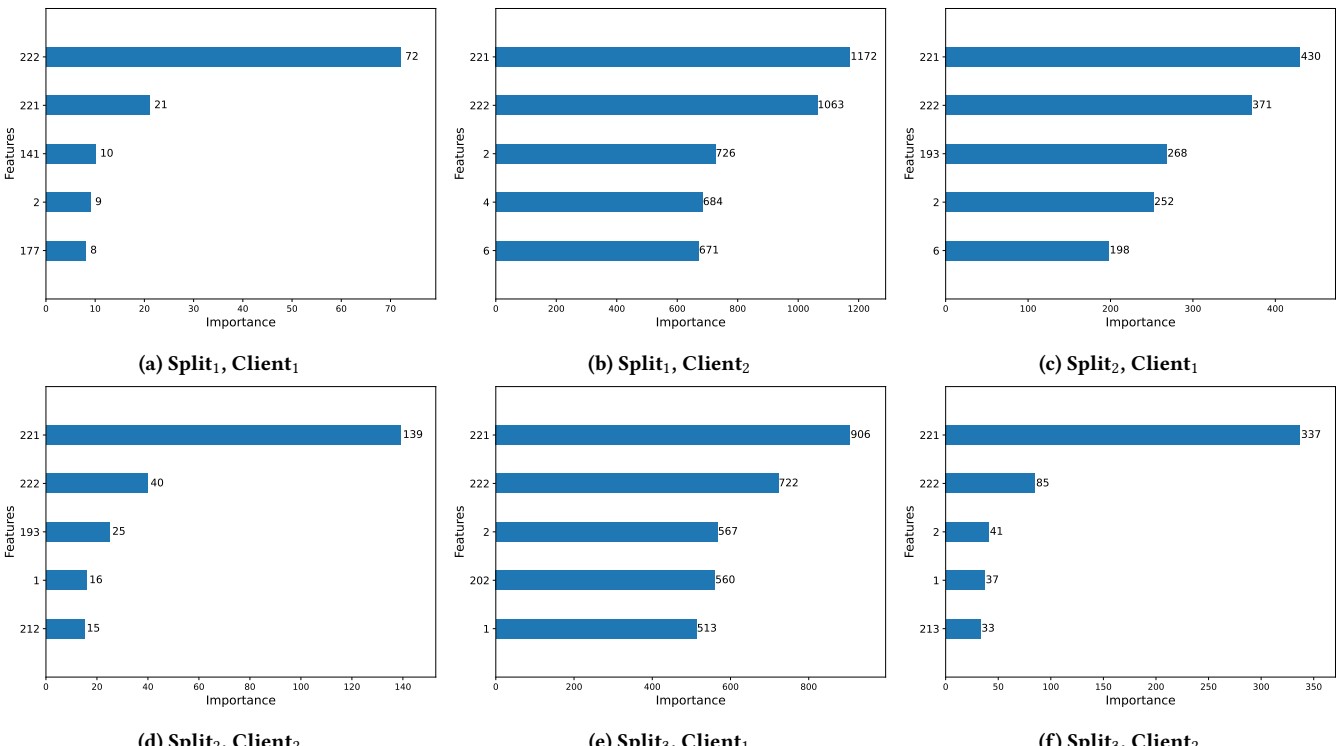

**Figure 3: Feature importance plots for the models trained using the model stacking approach**

datasets, the stacked model exploits their specializations in different aspects of the data, enhancing its effectiveness and generalization capabilities. The feature importance plots in Figure 3 provide an objective means to evaluate the meta-model's ability to effectively utilize the scores of the base models. However, a key limitation of this approach is that the dataset used to train the meta-model is not drawn from the same distribution as the test set, which may impact its generalization capabilities.

*Experiment repeatability.* To ensure repeatability of our experiment, in Table 6 we provide the hyper-parameters of the models trained on the local clients. The experimental setting is described in Subsection 5.1.

## 6  CONCLUSION AND FUTURE WORK

This study has explored strategies for addressing non-IID characteristics in Learning to Rank datasets. It has investigated effective strategies for partitioning a Learning to Rank dataset into two non-IID subsets and has examined methodologies for merging regression tree forests trained on the same non-IID datasets, aiming to enhance the overall predictive power of the final model. The findings indicate that leveraging dataset features for partitioning an LTR dataset into non-IID subsets can be effective, although, in real-world scenarios, pre-existing data partitioning may make this step unnecessary. In such cases, the focus shifts to the methods for combining LTR models. Results highlight that, particularly when a small amount of data is available for the merging task, the effectiveness of the combined model typically surpasses the baseline, improving overall predictive capability.

**Table 6: Hyperparameters of the models trained on the local clients**

|  |  | $M_1$ | $M_2$ |
|---|---|---|---|
| $Split_1$ | Learning rate | 0.05 | 0.05 |
|  | Number of leaves | 170 | 237 |
|  | Min data in leaf | 375 | 65 |
|  | Min sum of Hessian in leaf | 1 | 1 |
| $Split_2$ | Learning rate | 0.05 | 0.05 |
|  | Number of leaves | 193 | 150 |
|  | Min data in leaf | 20 | 40 |
|  | Min sum of Hessian in leaf | 1 | 1 |
| $Split_3$ | Learning rate | 0.05 | 0.05 |
|  | Number of leaves | 225 | 77 |
|  | Min data in leaf | 60 | 170 |
|  | Min sum of Hessian in leaf | 1 | 5 |

However, this study has certain limitations. It was tested only on one LTR dataset, and while many others are available, they may require different partitioning strategies. The absence of feature details in certain datasets, like the Yahoo! Learning to Rank Challenge dataset[6], poses challenges for our approach. The Microsoft LTR datasets[7] [18], in contrast, lack a document category feature. Consequently, exploring alternative features becomes necessary before attempting dataset partitioning. For instance, PageRank could serve as a viable option, given that documents with low PageRank values may prioritize lexical features. However, it's important to note that an ideal scenario would involve the availability of a LTR dataset already pre-partitioned into non-IID subsets.

Another limitation lies in the approach that selects which ranker to use at runtime. The selection phase needs improvement. A potential solution involves training a binary classifier to choose the ranker based on input features and selecting the model that obtains the highest NDCG@10 for the specific query. However, this approach would necessitate new data for training, altering the scenario of the original approach.

Future research directions include applying the methods to real-world scenarios and extending the LTR model merging to frameworks beyond that of Gradient Boosted Regression Trees, such as Neural Networks. These advancements could further enhance the effectiveness of the proposed strategies.

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
