# OpenReview forum: "Learning to Rank for Non Independent and Identically Distributed Datasets"
_ACM.org/SIGIR/ICTIR/2024/Conference — ICTIR 2024_

### Official Review · Reviewer_rU9M · 2024-05-16

**Rating:** -1
**Confidence:** 4

**Objective Part Of Review:**

Starting from introduction and abstract, this paper is clear in describing the problem and motivation. The paper aims to train a ranker in federated setting while keeping the data in each node locally. The motivation for this problem is clear as it is more privacy-friendly. The paper also explains that data in local nodes can differ with each other in terms of distribution and size and they aim to address the problem for non-IID data of type 1 and type 4. Since the publicly available data for this problem does not exists, the authors propose to generate data to evaluate the effectiveness of their method. In the evaluation section, the paper provides insights on the generated datasets, and investigates different strategies used for generating datasets  which can be relevant for problem evaluation. The strategies consider also extreme cases of dataset generation  for evaluation.

While the motivation and experiments are clear, the paper lacks explaining one primary point. As mentioned in section 3, the authors use Istella-S dataset and they explain that they assign categories to queries and later employ them for generating datasets. However, it is not clear what are these categories? in the dataset description in 3. they do not mention the categories, and in official Istella-S dataset description (https://istella.ai/data/letor-dataset/) there is no mention of a category attribute. Apart from this issue, the paper mentions that in [26] the authors proposes practical strategies for generating non-IID data, and since [26] considers search intent simulated for TREC web track queries which has a small size, they use Istella-S. Given the first research, the paper misses explaining if the strategies this paper suggests differ from the previous work or not.

**Subjective Part Of Review:**

Overall, this paper is organized and defines the problem clearly. The paper addresses an interesting problem which can eventually improve the privacy of training ranking models in a federated setting. The idea is simple and interesting. In terms of the results, the improvements are achieved, however; the paper only considers a two node setup in the evaluation, and given the analysis of generated datasets in terms of categories and partitioning strategies, the effectiveness of the solution may be less pronounced in a real-world setting.

---

### Official Review · Reviewer_2488 · 2024-05-17

**Rating:** 1
**Confidence:** 4

**Objective Part Of Review:**

The authors tackle the problem of federated learning to rank (LTR) and propose 1) A novel non-IID LTR data generation method and 2) a Linear model combination method for the federated LTR. Empirical results on public LTR datasets indicate that the proposed method works in practice.

The paper is very well-written, and the contributions are presented intuitively. The problem of federated LTR is very timely, given the recent laws around privacy, which will make it hard to keep customer data in a centralized platform. Online platforms relying on LTR can then apply federated LTR as a workaround to train their search or recommender systems.

**Subjective Part Of Review:**

- The paper is very well-written and easy to read.
- The proposed method is original and relevant to the LTR community, given the recent laws around privacy.
- Authors present a federated LTR method, based on a linear combination for the LambdaMART, which is SOTA in LTR and also the most widely used LTR method in industry.
- Empirical results are encouraging.

---

### Official Review · Reviewer_wWeP · 2024-05-22

**Rating:** 0
**Confidence:** 4

**Objective Part Of Review:**

The paper studies LTR on non-IID training data.

There are some strengths:

- The general problem of supervised ranking optimization in a federated search setup is of relevance and interest.  This follows popular privacy preserving literature on sharing models rather than data.

- The paper extends earlier research to a larger dataset Istella-S, with far more queries (although a low number of documents per query), with automatic categories assigned, simulated splits on the two most popular categories.

- The paper experiments with linear combination and model stacking approaches to combine the models trained on each split.

There are some limitations:

- While the experiment is very carefully constructed, the experiments are limited (as acknowledged in the conclusion section): one data set, internal comparison of two approaches, and no way to validate the setup to real-world scenarios.

- The "can be trivially extended to a greater number of clients" is perhaps true in some sense, but real world federated search over many clients feels far more challenging than over a pair clients.

- The way of assigning document/query categories is a key factor, as it drives the particular splits.  While this is a pragmatic choice, more discussion on how this may (or may not) have affected the outcomes would be welcome.

**Subjective Part Of Review:**

Minor:

- Although well written, it takes quite some effort to follow the exact logic of the splits (and trace the experimental setup).

- As the data set is one of the main contributions, replicating some of the earlier experiments on different data may have more clearly demonstrated the differences.

---

### Official Review · Reviewer_vAD1 · 2024-05-23

**Rating:** -1
**Confidence:** 5

**Objective Part Of Review:**

This paper consider the problem of learning a LTR ranker federatively across clients when data is distributed in a non-iid manner.

The paper is overall well written, and it consider an interesting, emerging and overlooked problem. It also deals with the challenge of not having datasets that can easily support empirical evaluation.

However, I have some fundamental concerns with the paper, of which concerns 2, 3 and 4 are the major concerns affecting my recommendation.

1) The focus is on federatively learning a ranker among n clients. The paper however empirically considers only 2 rankers in its evaluation, which is very limiting. The IStella dataset used here possibly allowed for more splits, so it is unclear why more clients weren't investigated

2) The paper builds upon the work of [26], which is different from this paper as they considered online LTR (so, click data, and iterative learning process), while here standard offline, label-based LTR is considered. In particular, the paper relies on the taxonomy of IID data put forward in [26], and it is said it consider type 1 data. However, this is not fully correct -- in fact, it is a bit of a misrepresentation. For type 1 in [26] the same queries are present in the data splits, but with a different intent in each split (thus resulting in different relevance labels given to the same query-doc pair across different splits). In this ICTIR paper, instead, the authors put into the two splits different queries: the queries that have documents (or the majority of documents) assigned to different categories. This is a major deviation; it also does not represent a clear non-IID setting: the distributions might be different because of topics, but there is no guarantee of this to be the case -- while it was in the setting of [26].

3) The paper correctly identifies the absence of a dataset for LTR with non-IID splits that could be used to inform and validate research in this area; this was also something cried out loud in [26]. The authors of this ICTIR paper decide not to use the dataset used in [26] because that only contained 200 queries -- this is a very valid concern/limitation of [26]. However, looking at Table 2, we then discover that also the dataset used here contains few queries. For example, in the exclusive partitioning setting (which is the most appropriate setting to consider according to the authors), queries for train can be as little as 239, and for testing as little as 25! So, somewhat, the concerned regarding using [26]'s data methodology don't seem to add up when considering the alternative experimental setup here.
Also, why not considering at least a LOOCV setup?

4) I wonder about the practicality of the proposed solutions. The methods proposed in the paper to merge the local rankers require the availability of the D_merge set, local to each client. The fact that D_merge is local to each client is good: there is no need to share this data, thus satisfying the typical requirements of a federated system like this. However, the clients do need to have prior knowledge of the data used for train, and importantly of that used for testing, in order to form D_merge. In other words, D_merge is formed in an artificial way. How would I be expected to form this data partition in practice?

5) The federated setup considered in the paper is somewhat unusual. Yes, as mentioned by the authors, they do not consider repeated/iterative learning over epochs of federated learning, as done commonly in FL papers (including in the FOLTR literature because of the online learning process). But, additionally, the setup is also different because there is no global model being learnt and then distributed to clients (and possibly adapted locally): instead each local rankers learns a new local ranker, biased to the local characteristics, based on the local ranker itself and the ranker of the other client. This setup is uncommon in FL, though related to personalisation/localisation in FL models, and to peer-to-peer FL (i.e. where models are learnt without a central server). I could not see these items being adequately teased out in the related works/methods/discussions of the paper.

Other minor comments:

M1) In section 2, [15] is referred to as (the only) example of federated online LTR technique. [15] is indeed the first (of two) that have come out in this area, but it is also quite problematic: https://link.springer.com/chapter/10.1007/978-3-030-72240-1_10 shows the original findings might not be reproducible, and do not generalise outside the original dataset. Also, the current SOTA in this area is https://dl.acm.org/doi/abs/10.1145/3471158.3472236

M2) There is no need to use the word "metric" in "NDCG@10/MAP metric".

M3) It was unclear to me what model was used for the merging function for the model stacking method. The paper says "a new simple model" (section 4)... what model is it? Is it a linear model? a tree model? a shallow neural model?

M4) in section 5.2, the text under table 3 says that "the local models exhibit the desired characteristics". I am not sure these are desired characteristics (not by the user for sure... perhaps by the authors of the paper): I would say they show the expected characteristics of a non-IID setup.

**Subjective Part Of Review:**

I struggle with this paper in that I have a high regard for the work done here, and the challenging problem the authors faced in building an evaluation setup for their method -- a challenge that is admittedly a big roadblock for research in this area.
So, I am looking at this paper very favourably from the onset. In addition, the paper is well written and often insightful.

On the other hand, however, there are major methodological concerns (expressed above).
ICTIR traditionally has lesser of a focus on complete evaluation -- so I concede that point 1 above could be discarded. Point 3 above is also related to the empirical experiments; however, to me, it displays an incongruence I would have preferred not seeing, or at least being acknowledged. Point 2 however is, I feel, an issue that strikes to the core tenets of the paper.

In other words: I would like to accept this paper; however I have difficulties in doing so confidently in its current version.

---

### Meta-Review · Area_Chair_usfY · 2024-05-30

**Recommendation:** Accept (Oral)
**Confidence:** 3

**Metareview:**

The reviewers agreed that the paper is well-presented and that it addresses an interesting and relevant task. They raised several major concerns, specifically regarding the data used in the experiments and the applicability of the approach, that require changes and additions to the paper:
1. The data splits used in the paper are different from those used in [26], and additional discussion and justification are needed.
2. The authors did not use the dataset from [26] for experiments and did not replicate some earlier baselines on their dataset.
3. Limited empirical exploration: the authors only used one dataset and two splits.
4. There are concerns about the practicality of the proposed approach.

Given the theoretical nature of the ICTIR conference, providing an informative discussion on points 3 and 4 above should help ease the reviewers' concerns. Points 1 and 2 are the most critical ones. Considering the interesting theoretical contribution of the work and the fact that the authors address a task for which there is no large enough dataset for experiments, it may be reasonable to overlook these issues. That said, the authors are asked to explain the design choices made when creating their dataset better and present the performance of their approach on the dataset proposed in [26], even if it is relatively small.

I believe the other concerns raised by the reviewers, including missing details about the categories assigned to the documents in the dataset and the model of the merging function, can be rather easily addressed in the final version of the paper.